# Cathepsin B-Cleavable Polymeric Photosensitizer Prodrug for Selective Photodynamic Therapy: In Vitro Studies

**DOI:** 10.3390/ph15050564

**Published:** 2022-04-30

**Authors:** Manish Jain, Jordan Bouilloux, Ines Borrego, Stéphane Cook, Hubert van den Bergh, Norbert Lange, Georges Wagnieres, Marie-Noelle Giraud

**Affiliations:** 1Department EMC, Faculty of Sciences and Medicine, University of Fribourg, CH-1700 Fribourg, Switzerland; manishjain.cdri@gmail.com (M.J.); ines.borrego@unifr.ch (I.B.); stephane.cook@h-fr.ch (S.C.); 2Pharmacology Division, University Institute of Pharmaceutical Sciences (UIPS), Panjab University, Chandigarh 160014, India; 3School of Pharmaceutical Sciences, Laboratory of Pharmaceutical Technology, Institute of Pharmaceutical Sciences of Western Switzerland, University of Geneva, Rue Michel-Servet 1, CH-1211 Genève, Switzerland; jordan.bouilloux@unige.ch (J.B.); norbert.lange@unige.ch (N.L.); 4HFR Hôpital Fribourgeois, CH-1708 Fribourg, Switzerland; 5Medical Photonics Group, LCOM-ISIC, Swiss Federal Institute of Technology (EPFL), CH-1015 Lausanne, Switzerland; hubert.vandenbergh@epfl.ch; 6Laboratory for Functional and Metabolic Imaging, LIFMET, Swiss Federal Institute of Technology (EPFL), CH-1105 Lausanne, Switzerland; georges.wagnieres@epfl.ch

**Keywords:** cathepsin B, photosensitizer, photodynamic therapy, prodrug

## Abstract

Cathepsin B is a lysosomal cysteine protease that plays an important role in cancer, atherosclerosis, and other inflammatory diseases. The suppression of cathepsin B can inhibit tumor growth. The overexpression of cathepsin B can be used for the imaging and photodynamic therapy (PDT) of cancer. PDT targeting of cathepsin B may have a significant potential for selective destruction of cells with high cathepsin B activity. We synthesized a cathepsin B-cleavable polymeric photosensitizer prodrug (CTSB-PPP) that releases pheophorbide a (Pha), an efficient photosensitizer upon activation with cathepsin B. We determined the concentration dependant uptake in vitro, the safety, and subsequent PDT-induced toxicity of CTSB-PPP, and ROS production. CTSB-PPP was cleaved in bone marrow cells (BMCs), which express a high cathepsin B level. We showed that the intracellular fluorescence of Pha increased with increasing doses (3–48 µM) and exerted significant dark toxicity above 12 µM, as assessed by MTT assay. However, 6 µM showed no toxicity on cell viability and ex vivo vascular function. Time-dependent studies revealed that cellular accumulation of CTSB-PPP (6 µM) peaked at 60 min of treatment. PDT (light dose: 0–100 J/cm^2^, fluence rate: 100 mW/cm^2^) was applied after CTSB-PPP treatment (6 µM for 60 min) using a special frontal light diffuser coupled to a diode laser (671 nm). PDT resulted in a light dose-dependent reduction in the viability of BMCs and was associated with an increased intracellular ROS generation. Fluorescence and ROS generation was significantly reduced when the BMCs were pre-treated with E64-d, a cysteine protease inhibitor. In conclusion, we provide evidence that CTSB-PPP showed no dark toxicity at low concentrations. This probe could be utilized as a potential imaging agent to identify cells or tissues with cathepsin B activity. CTSB-PPP-based PDT results in effective cytotoxicity and thus, holds great promise as a therapeutic agent for achieving the selective destruction of cells with high cathepsin B activity.

## 1. Introduction

Proteases such as cathepsins, or matrix metalloproteinases, are upregulated in various pathologies and are potential therapeutic targets. In particular, cathepsin B (CTSB) overexpression is associated with metastatic cancer [1], atherosclerotic plaques [2], neurodegenerative diseases [3], and other inflammatory diseases, including COVID-19 [4]. CTSB is a lysosomal cysteine protease involved in the degradation of proteins and organelles, antigen presentation, and autophagy [5]. In physiological conditions, it is highly regulated. The dysregulation of cathepsin B causes cells to acquire pathological phenotypes. In recent years, strategies have been proposed for the selective targeting of cysteine proteases, including the use of synthetic small molecule inhibitors or antibodies directed against cathepsins [6,7,8,9]. Nevertheless, most synthetic inhibitors usually have low bioavailability and often exert reactivity toward off-target proteins [10].

Photodynamic therapy (PDT) utilizes photosensitizer (PS), which upon activation via a light of a specific wavelength, generates cytotoxic ROS [11] that leads to cell death via the loss of mitochondrial membrane potential, lipid peroxidation, or protein denaturation of cellular membranes and organelles [12,13]. PDT presents a promising alternative for pathologies associated with vascular abnormalities, inflammatory processes, and the development of neoplasia [14]. To selectively target the cells overexpressing CTSB, we developed a cathepsin B-cleavable polymeric PS prodrug (CTSB-PPP). CTSB-PPP consists of a poly-l-lysine backbone, to which multiple copies of the PS ‘pheophorbide a’ (Pha) units are tethered via a cathepsin B-cleavable short peptide sequence.

Pha is a chlorin-based PS that has demonstrated a beneficial effect against leukaemia, colon cancer, uterine carcinosarcoma, and rheumatoid arthritis [15,16]. Several photosensitizers have been approved for clinical applications or under clinical trials [17]. Photofrin^®^ (Axcan Pharma, Canada) is the first type to be clinically approved photosensitizer for the treatment of cancer. The second generation photosensitizers, such as Temoporfin (Foscan^®^, Biolitec, Germany), Motexafin lutetium, Palladium bacteriopheophorbide (TOOKAD^®^, Negma-Lerads), Purlytin^®^, Verteporfin (Visudyne^®^, Novartis, Switzerland), and Talaporfin (Laserphyrin^®^, Meiji Seika, Japan) are clinically approved or under-clinical trials. However, PDT using a conventional photosensitizer has disadvantages such as photosensitive side effects, low light penetration depth, inconvenience, and relatively high cost. In contrast, Pha is a promising photosensitizer for PDT that meets most of the criteria that a good photosensitizer should satisfy [18]. In a comparative study, Djalil A.D. et. al. [19] reported that the drug concentration that causes a 50% reduction in cell viability (the LC50) of pheophorbide-a was much lower (<5µM) than that of Protoporphyrin IX, suggesting a higher dose of the former. When used for diagnosis, in vitro studies demonstrated that Pha showed almost five times more fluorescence brightness than chlorin-e6 [20]. Pha thus provides a better signal-to-noise ratio with a better detection limit, which is a critical factor required for imaging probes.

Fluorescence emission and phototoxicity of the Pha units in the CTSB-PPP molecule are quenched via the “switch-off state” due to the close proximity of the rather lipophilic Pha molecules. Photoexcitation of CTSB-PPP thus leads to exciplex formation and quenching via internal conversion. In the target tissue, after proteolytic cleavage of the peptide sequence, the inactive PS becomes active, increasing the distance between the Pha molecules, i.e., the “switch-on state” [10,21,22,23,24]. These active PS can fluoresce upon activation via light of an appropriate wavelength (~670 nm in case of Pha), in the presence of molecular oxygen, leading to the generation of reactive oxygen species (ROS) that locally destroys cells over short diffusion distances [25].

The present work aims at (1) synthesizing a novel Pha-based CTSB-PPP, (2) investigating its safety using two different models, one in vitro and one ex vivo, and (3) assessing the in vitro efficacy of CTSB-PPP-based PDT. For in vitro studies, bone marrow cells (BMC) were chosen as a model, as they express a high content of CTSB.

## 2. Materials and Methods

### 2.1. Chemicals

Penicillin/Streptomycin, Trypsin-EDTA, Methylene tetrazolium (MTT), dihydroethidium (DHE), 1-ethyl-(3-dimethylaminopropyl)carbodiimide hydrochloride (EDC), 4-dimethylaminopyridine (DMAP), *N*-hydroxysuccinimide (NHS), poly-l-Lysine (PLL, 25 kDa), and other fine chemicals were obtained from Sigma (Buchs, Switzerland). Antibodies including CD90, CD44, CD29, and E-64d were purchased from Abcam (Cambridge, UK). Cathepsin B polyclonal antibody (USCPAC964Mu01) was purchased from Chemie Brunschwig (Basel, Switzerland). Dulbecco’s Modified Eagle Medium (DMEM) was purchased from vWR International (Nyon, Switzerland). Pheophorbide a (Pha) was purchased from Frontier Scientific (Logan, UT, USA). GAGRRAAG peptide was obtained from Caslo Laboratory (Lyngby, Denmark). mPEG-NHS 20 kDa was purchased from NANOCS (New York, NY, USA). *O*-(7-azabenzotriazole-1-yl)-*N*,*N*,*N*′N′-tetramethyluronium hexafluorophosphate (HATU) was obtained from GenScript Corporation (Piscataway, NJ, USA). Trifluoroacetic acid (TFA, 99%), *N*,*N*′-diisopropylethylamine (DIPEA), *N*,*N*′-dimethylformamide (DMF), and dimethylsulfoxide (DMSO) were obtained from Acros Organics (Thermo Fisher Scientific, Wohlen, Switzerland). *N*-succinimidyl(1-methyl-3pyridinio)formate iodide was synthesized according to the literature [26]. Pha-NHS was prepared as described previously [27].

### 2.2. Synthesis and Characterization of CTSB-PPP

#### 2.2.1. Synthesis of Pha-CTSB

Pha-NHS (20 mg, 2.9 × 10^−5^ moles), cathepsin B-cleavable peptide sequence, i.e., GAGRRAAG (TFA salt) (48 mg, 5.8 × 10^−5^ moles) [28] and DIPEA (20.2 μL, 1.2 × 10^−4^ moles) were stirred in DMF (10 mL) under argon in the dark at room temperature overnight. H_2_O (10 mL) and TFA (30 μL) were added to reach pH = 4. Solvents were removed under reduced pressure, and the crude product was purified by semi-preparative reversed-phase high-performance liquid chromatography (RP-HPLC) on an automatic PuriFlash 4100 instrumentation with an Interchim Soft version 5.0x software from Interchim (Montluçon, France). Separation was achieved on a Nucleodur^®^ C_18_ HTec column (762,556.210; 5 μm, 21 × 250 mm) from Macheray-Nagel (Oensingen, Switzerland) with one of the following systems: 9 mL/min, 30–100% solvent B in 30 min (with H_2_O + 0.01% TFA as solvent A and ACN + 0.01% TFA as solvent B), detection at 210, 313 and 410 nm. The final product was then lyophilized to give a green solid (33.2 mg, 88.8% yield). ^1^H NMR (300 MHz, DMSO-*d*_6_) δ 12.60 (s, 1H), 9.82 (s, 1H), 9.48 (s, 1H), 8.90 (d, *J* = 16.0 Hz, 1H), 8.33–7.85 (m, 10H), 7.44 (d, *J* = 5.6 Hz, 2H), 6.48–6.32 (m, 2H), 6.24 (dd, *J* = 11.6, 1.6 Hz, 1H), 4.57 (q, *J* = 6.8 Hz, 1H), 4.26 (dq, *J* = 18.1, 7.1 Hz, 6H), 4.04 (d, *J* = 8.2 Hz, 1H), 3.87–3.56 (m, 17H), 3.43 (d, *J* = 11.3 Hz, 4H), 3.26 (d, *J* = 12.3 Hz, 7H), 3.11–3.01 (m, 5H), 2.48–2.39 (m, 1H), 2.36–2.25 (m, 1H), 2.22 (s, 6H), 2.09 (d, *J* = 12.1 Hz, 1H), 1.77 (d, *J* = 7.1 Hz, 3H), 1.63 (h, *J* = 7.9, 6.4 Hz, 6H), 1.46 (s, 9H), 1.18 (ddd, *J* = 14.9, 6.7, 3.0 Hz, 11H), 0.47 (s, 1H), −1.79 (s, 1H). ESI-MS: *m/z* 1290.3 [M + H]^+^; 645.8 [M + 2H]^2+^.

#### 2.2.2. Synthesis of CTSB-PPP

PLL (25 kDa, 9.42 mg, 3.9 × 10^−7^ moles, 4.5 × 10^−5^ moles of -NH_2_ functions) was added to a solution of Pha-CTSB (16.3 mg, 1.26 × 10^−5^ moles, 28% of the -NH_2_ functions of the PLL backbone) and HATU (6.54 mg, 1.72 × 10^−5^ moles, 1.3 equivalents of the Pha-CTSB to be activated) in DMSO (3 mL). Following activation, DIPEA (45 µL, 2.58 × 10^−4^ moles) was added to the mixture, which was then stirred under argon, and the reaction was allowed to proceed in the dark overnight. mPEG-NHS (20 kDa, 9.93 mg, 4.95 × 10^−5^ moles, 1% of the -NH_2_ functions of the PLL backbone, 1.1 equivalent) and DIPEA (15 µL) in DMSO (1.5 mL) were added to the mixture, and the reaction was allowed to proceed in the dark overnight. Remaining free -NH_2_ functions of the PLL backbone were capped with *N*-succinimidyl(1-methyl-3pyridinio)formate iodide (12.75 mg, 3.51 × 10^−5^ moles, 71%, 1.1 equivalent) and DIPEA (15 µL) in DMSO (1.2 mL), and the reaction was allowed to proceed in the dark overnight. The reaction was quenched by adding a mixture of water and TFA to reach pH = 3. Solvents were removed under reduced pressure, and the crude was purified by size exclusion chromatography (SEC) using a Sephacryl^TM^ S-100 column (Amersham Biosciences, Otelfingen, Switzerland) and H_2_O/ACN/TFA (70:30:0.01) as the eluting solvent (1.2 mL/min). The final product was lyophilized to give a green solid (31.3 mg, 80.4% yield).

### 2.3. Quenching Efficiency

The corresponding polymer carrying only 1% of peptide-Pha was prepared to assess the quenching efficiency. To this end, PLL (25 kDa, 22.93 mg, 9.17 × 10^−7^ moles, 1.10 × 10^−4^ moles of -NH_2_ functions) was added to a solution of Pha-CTSB (1.41 mg, 1.09 × 10^−6^ moles, 1% of the -NH_2_ functions of the PLL backbone) and HATU (0.58 mg, 1.52 × 10^−6^ moles, 1.3 equivalents of the Pha-CTSB to be activated) in DMSO (1 mL). DIPEA (60 µL, 3.44 × 10^−4^ moles) was then added to the mixture, which was then stirred under argon, and the reaction was allowed to proceed in the dark overnight. mPEG-NHS (20 kDa, 24.16 mg, 1.21 × 10^−6^ moles, 1% of the -NH_2_ functions of the PLL backbone, 1.1 equivalent) and DIPEA (10 µL, 5.74 × 10^−5^ moles) in DMSO (1 mL) were added to the mixture, and the reaction was allowed to proceed in the dark overnight. The remaining free -NH_2_ functions of the PLL backbone were capped with *N*-succinimidyl(1-methyl-3pyridinio)formate iodide (42.91 mg, 1.18 × 10^−4^ moles, 98%, 1.1 equivalent) and DIPEA (10 µL, 5.74 × 10^−5^ moles) in DMSO (1 mL), and the reaction was allowed to proceed in the dark overnight. The reaction was quenched by adding a mixture of water and TFA to reach pH = 3. Solvents were removed under reduced pressure, and the crude was purified by SEC using a Sephacryl^TM^ S-100 column (Amersham Biosciences, Otelfingen, Switzerland) and H_2_O/ACN/TFA (70:30:0.01) as the eluting solvent (1.2 mL/min). The final product was then lyophilized to give a green solid.

### 2.4. Spectral Characterization

The absorbance and emission spectra of CTSB-PPP and its reference conjugate grafted with 1% Pha-peptide (3 μM Pha equivalent) were measured at 37 °C using a Biotek Multiplate reader (Synergy 2, Biotek, Winooski, VT, USA), setting the excitation at 400 nm and emission at 670 nm. The gain was set to 100, and the slit width was 9, with an accumulative number of 5. Five samples of both conjugates were prepared and measured in the 96-well quartz plate. The fluorescence quenching factor (x-fold decrease in background subtracted fluorescence at the 670 nm emission maximum) was calculated with respect to the non-quenched reference conjugate.

### 2.5. Animal Experimentation

Seven-week-old male Lewis rats were used in the present study. The animals were obtained from Charles River Laboratories, France. All the procedures involving animal experimentation were performed in compliance with the European Convention on Animal Care in accordance with the Swiss Animal Protection Law after obtaining permission from the State Veterinary Office, Fribourg, and approved by the Swiss Federal Veterinary Office, Switzerland (FR 2013-35). The animals were housed in individually ventilated cages and were maintained on a chow diet, water ad libitum, and 12 h/12 h light-dark cycle.

### 2.6. Cell Culture Conditions

BMCs were isolated from the bone marrow of a male Lewis rat and cultured in 75 cm^2^ containing Dulbecco’s Modified Eagle Medium supplemented with 10% Fetal Bovine Serum and 1% Penicillin/Streptomycin. BMCs were selected upon adherence, and the culture medium was removed after 2 days to remove non-adherent cells. The medium was subsequently replaced after 3 days, and the BMCs were allowed to grow for 8–10 days to reach confluence [29]. The cells were harvested using Trypsin-EDTA and were sub-cultured at a density of 2 .1 × 10^6^ cells in a 75 cm^2^ flask. BMCs were characterized by flow cytometry for the presence of surface markers CD90, CD44, and CD29 using specific antibodies [30]. All experiments were performed on confluent cells within passages of 3–4.

### 2.7. In Vitro Activation of CTSB-PPP

BMCs were seeded into 48-well plates at a density of 10^5^ cells per well and then cultured in the medium for 24 h to attach. To evaluate the dose-dependent cellular activation of CTSB-PPP, cells were washed and treated overnight, with or without E-64d (50 µM). Cells were washed with PBS, and solutions of CTSB-PPP in DMEM at final concentrations of 6, 12, 24, or 48 µM of Pha equivalents were added for 60 min. In another set of experiments, cells were treated with 6 µM CTSB-PPP for 10 min, 20 min, 60 min, or 24 h. Following treatment, cells were immediately washed twice. The cellular localization of CTSB-PPP was visualized using a confocal microscope (Leica TCS SP5 DMI6000) equipped with a Leica plan apo 20× (numerical aperture 0.7) dry objective. Red fluorescence of the PS and corresponding bright-field images were collected in the fluorescence emission range of 670–700 nm using a 405 nm diode laser for excitation [31,32]. Pictures were taken using Leica Application Suite Advanced Fluorescence (LASAF) software (Leica Microsystems). Fluorescence intensity per cell was quantified using the ImageJ, software as previously described [33,34].

### 2.8. MTT Assay

To assess the dark toxicity of CTSB-PPP, BMCs were seeded into 96-well plates at a density of 2 × 10^3^ cells/well and were allowed to adhere overnight. Cells were then treated with the indicated concentration of CTSB-PPP, in the presence or absence of E-64d, for 60 min. Subsequently, cells were washed and returned to the incubator, and viability was evaluated after 24 h by MTT assay. Briefly, cells were incubated with MTT (20 μL, 5 mg/mL in PBS) at 37 °C for 4 h. After incubation, the formazan product was dissolved in DMSO, and absorbance was read at 570 nm using an ELISA plate reader. The percentage of cell metabolic activity as an indicator of cell viability was calculated with respect to control samples as: (% cell viability = (mean OD value of the irradiated cells/mean OD value of the non-irradiated cells) × 100) [35,36].

### 2.9. Light Source and PDT

PDT efficacy was determined in BMCs treated with or without CTSB-PPP (6 µM). Immediately after one hour of treatment, cells were illuminated with red light using a frontal light diffuser (FD-1, Medlight SA, Switzerland) coupled to a 1W Diode laser (Frankfurt Laser Company, Germany) emitting at 671 nm. This diffuser has a graded-index lens at its distal end and two perpendicular mode scramblers to give nearly perfect homogenous light intensity distinguished over a large circular area. Laser power emitted by the frontal light diffuser was calibrated with a power meter (Spectra-Physics Newport; model 407A) according to a procedure described in detail by Borle et al. [37]. The light diffuser was placed 1 cm away from the bottom of the well, and total light doses ranging from 0 J/cm^2^ to 200 J/cm^2^ were applied at a fluence rate of 100 mW/cm^2^ [23,38]. After PDT, MTT assays were performed to assess the cell viability, as described above.

### 2.10. Detection of ROS

Cellular ROS was detected by DHE staining. Immediately after PDT, cells were rinsed with PBS and incubated with DHE (10 μM) in a dark, humidified chamber for 10 min at 37 °C. Cells were washed with PBS and counterstained with Hoechst (5 μg/mL) [39]. Fluorescent images were acquired with the same exposure time from different groups on a Nikon Ni-U microscope (Nikon, Tokyo, Japan). DHE fluorescence intensity normalized with the area was measured using ImageJ software and is expressed as DHE fluorescence intensity per µm^2^ (arbitrary units).

### 2.11. Assessment of Vascular Function

Aortic segments from C57BL/6J were cut into rings (2 mm in length) and mounted between two L-shaped hooks in a Multi-Myograph System (Model 720 MO, Danish Myo Technology A/S, Denmark). Myograph chambers were filled with Krebs bicarbonate buffer and were bubbled with 95% O_2_ and 5% CO_2_ at 37 °C. After equilibration, the aortic rings were exposed to KCl Krebs buffer (80 mM) to assess the maximum tissue contractility. Changes in isometric tension (Δ mN) were recorded with Lab Chart Pro v8.0.5 software (AD Instruments, UK). The vasoconstriction response to phenylephrine (PE) was determined as the increase in force (Δ mN) from the baseline upon cumulative addition of PE (1 nM–100 µM). The presence of a functional endothelium was verified by the occurrence of significant relaxation to ACh (3 nM–300 µM) in PE (1 µM) precontracted rings [40]. Aortic rings were subsequently treated with CTSB-PPP (6 µM for 1 h), and after washing, another set of PE and ACh-mediated responses were taken. Finally, tissue contractility and viability were assessed by exposing the rings to KCl Krebs buffer (80 mM) [41,42].

### 2.12. Statistical Analysis

Experimental values in the results are presented as means with their standard deviation. All experiments were performed at least three times. The statistical significance of the difference between means was assessed using the unpaired two-tailed Student’s *t*-test (for the comparison of two groups) or by one-way ANOVA followed by Bonferroni’s multiple-comparisons tests (parametric data of more than two groups). *p* < 0.05 was considered as statistically significant. All statistical analyses were performed with the GraphPad Prism 5.0 program (GraphPad, Inc., San Diego, CA, USA).

## 3. Results

### 3.1. Preparation and Characterization of CTSB-PPP

The prodrug CTSB-PPP was synthesized as described above. CTSB-PPP consisted of multiple copies of the PS Pha attached to a poly-l-lysine backbone via a short, cathepsin B-cleavable peptide sequence (Figure 1). CTSB-PPP showed the typical absorption spectra from multiple Pha units (Figure 2A), in agreement with the results of our previously reported polymeric PS prodrugs [23,27,32,43]. PS loading of 28% of the lysine side chains per polymer resulted in the efficient quenching of fluorescence emission near 670 nm. As shown in Figure 2B, CTSB-PPP was 146 times less fluorescent than the equimolar amount of Pha from the non-quenched reference conjugate.

### 3.2. Cellular Uptake, Specificity, and Dark Toxicity of CTSB-PPP

After entering the cells, the peptide linker GAGRRAAG is cleaved by cathepsin B, releasing free Pha that can be detected via fluorescence imaging. First, the expression of cathepsin B in BMCs was confirmed by immunostaining (Appendix A). Then, the uptake efficiency of CTSB-PPP within BMCs was examined after one hour of treatment using confocal microscopy (Figure 3A). The presence of a Pha-derived red fluorescence signal in cells treated with CTSB-PPP reflected the activation of the probe within the cell. The quantification of the fluorescence intensity (FI) revealed a dose-dependent increase in the accumulation of CTSB-PPP. Compared to background auto-fluorescence from control cells (FI = 278 ± 38), FI was significantly higher in cells treated with CTSB-PPP at a 6 µM concentration (4174 ± 502) or 12 µM (4834 ± 906). A further significant increase in FI was noted by increasing the concentration to 24 µM (6841 ± 948). However, no further change in FI was noted in BMCs treated with 48 µM CTSB-PPP (7235 ± 1241) (Figure 3C, Appendix A). The safety of CTSB-PPP was also investigated in parallel by examining its dark cytotoxicity. BMCs were treated for 1 h, with or without 6, 12, or 24 µM CTSB-PPP. When compared to the control (100 ± 14.9%), CTSB-PPP treatment at 6 µM concentration did not exert any significant loss in cell viability (89.7 ± 19.5%), as determined by MTT assay. However, a significant reduction in cell viability was noted at higher concentrations of CTSB-PPP, i.e., 12 µM (71.7 ± 7.9%) and 24 µM (57.0 ± 15.4%) (Figure 3D). To define the specificity of CTSB-PPP, BMCs were treated with E-64d (an irreversible cell-permeable cysteine protease inhibitor) (Figure 3B). Pre-treatment with E-64d (50 µM) led to a significant reduction in FI of CTSB-PPP at 6 µM (2114 ± 705) or 12 µM concentration (2837 ± 395), as compared to their cells without E-64d (Figure 3C).

### 3.3. Time-Dependent CTSB-PPP Accumulation

To assess the PDT efficacy of CTSB-PPP, we first identified the optimal time required by CTSB-PPP to accumulate in BMCs. BMCs were treated with CTSB-PPP 6 µM (non-toxic dose) for different times, and FI was quantified on corresponding fluorescent images (Figure 4A). A sharp increase in FI, representing activation and accumulation of the CTSB-PPP probe, was noted as early as 10 min (1932 ± 496), which slightly increased at 20 min (2211 ± 222) and peaked at 60 min of treatment (3585 ± 505). Increasing the treatment time to 24 h resulted in a slight reduction in fluorescence (3045 ± 262) (Figure 4B, Appendix A).

### 3.4. Ex Vivo Toxicity

We assessed the effect of CTSB-PPP (6 µM) on vascular function using a Myogram. KCl induced maximal tissue contractility, dose-dependent phenylephrine (PE) induced contraction, and acetylcholine (ACh) induced relaxation were determined before and after CTSB-PPP treatment (6 µM for 1 h). The KCl response, PE induced maximal contraction, and Ach-induced endothelium-dependent relaxation were not altered after CTSB-PPP treatment (Figure 5A), suggesting that CTSB-PPP per se has no toxic effect on vascular function. We also confirmed the activation of CTSB-PPP ex vivo. As shown in Figure 5B, Pha was observed in the endothelium of the aortic ring.

### 3.5. CTSB-PPP Based PDT Efficacy

BMCs, treated with or without CTSB-PPP (6 µM for 1 h), were irradiated with a light dose of 0 to 100 J/cm^2^**.** Light doses up to 100 J/cm^2^ did not exert any significant change on cell viability; however, at 200 J/cm^2^, cell viability was reduced to 37.6 ± 2.4%, as compared to the control non-illuminated cells (100 ± 14.1%) (Figure 6A). A drastic reduction in cell viability was observed in cells treated with CTSB-PPP (6 µM for 1 h) and illuminated with 12.5 (54.1 ± 7.5%), 25 (51.2 ± 6.8%), 50 (35.2 ± 3.4%), or 100 J/cm^2^ (32.5 ± 3.4), as compared to non-illuminated cells treated with the same dose of CTSB-PPP (95.0 ± 5.1%) (Figure 6B).

### 3.6. PDT with CTSB-PPP Stimulated ROS Generation

It is well accepted that PDT utilizes light to excite a PS and generate cytotoxic ROS [14]. Therefore, we determined whether the phototoxic effect of CTSB-PPP was associated with ROS production. ROS levels were determined immediately after PDT using DHE staining (Figure 7A). DHE fluorescence intensity (arbitrary units) was not significantly changed in only CTSB-PPP-treated (46,194 ± 30,346) or only laser-irradiated groups (44,100 ± 35,834), as compared to the vehicle-treated control (22,370 ± 22,288). A significant increase in ROS levels was observed in CTSB-PPP photo-irradiated cells (20,2475 ± 32,952) compared to vehicle-treated control. Pre-treatment with E-64d (50 µM) led to a significant reduction in cellular ROS levels of CTSB-PPP photo-irradiated cells (86,686 ± 40,420) (Figure 7B, Appendix A). Original fluorescent photomicrograph of DHE staining, along with Hoechst staining, are presented in Appendix A.

## 4. Discussion

We developed a new hydrophilic PS prodrug to selectively target cathepsin B, which is highly upregulated in several pathologies, including immune disorders, cardiovascular diseases, and cancer. Our results demonstrated that the fluorescence of the prodrug was efficiently quenched in the prodrug’s native state. Intracellular activation of the prodrug in BMCs, following the selective cleavage of the peptide linker by cathepsin B, restores photoactivity of the Pha, which can be imaged and quantified by microscopy. The photo-activation of CTSB-PPP also resulted in the efficient reduction in BMCs viability. The probe can thus target upregulated cathepsin B protease activity and is, therefore, a very selective theranostic molecule.

In the current study, we have used the GAGRRAAG peptide sequence, which was designed from the AGRRAA sequence identified as a CTSB-specific substrate by Ruzza et al. [28]. Two G residues (at the amino- and carboxyl-end) were added for further chemical modification. Previously, we have also shown the proteolytic cleavability of the GAGRRAAG peptide linker using CTSB from a human placenta [24]. CTSB-sensitive PPP contains multiple copies of Pha as the PS. Pha is a product of chlorophyll breakdown and, upon photo-illumination at around 666 nm, it induces apoptosis and necrosis in cells via the generation of ROS and the release of cytochrome c [44,45]. Besides this known toxicological profile, the strong absorption band at 666 nm allows relatively deep tissue penetration, thus making it a suitable PS for our prodrug strategy. The Pha moieties are linked to a poly-l-lysine backbone via a peptide sequence. Previously, we have showed that the coupling of varying amounts of pheophorbide a-NHS ester to PLL results in the complete disappearance of the peak corresponding to pheophorbide a-NHS (λ_abs_ = 400 nm) and the appearance of a new, more polar, broad peak corresponding to PPP, thus providing conjugates with an estimated average of 1, 6, 12, 18, 24, and 30 PS units per polymer chain [27]. The fluorescence quenching factor increased as a function of the number of PS units in a polymer chain, revealing that at equimolar concentrations of the PS (equi-absorbant solutions at 667 nm), better fluorescence quenching is observed for higher PS loading [27]. Earlier, Ruzza et al. [28] showed that the specific sequence Arg-Arg-Ala-Ala exhibits the best selectivity for efficient enzymatic hydrolysis at lysosomal pH. Cleavage of the peptide linker by CTSB releases Pha, which could be detected at the excitation and emission maxima of 405 nm and 670 nm, respectively. In the present work, we have used 28% of PS-peptide loading per polymer chain [27,46], resulting in efficient fluorescence quenching.

The localization of a PS in sufficient concentration in a target tissue of interest is essential for the target’s fluorescence detection and its selective removal by photodynamic action. We used BMCs to assess intracellular activation, fark toxicity, and PDT-induced cytotoxicity of CTSB -PPP, as they are known to express cathepsin B [46,47]. Our results showed the homogeneous activation of CTSB-PPP, indicating the degradation of the peptide linker resulting in the restoration of fluorescence. CTSB-PPP demonstrated dose-dependent accumulation, where the maximum accumulation appeared after about 1 h. Both active and/or passive diffusion processes may account for this accumulation. A slight reduction in fluorescence intensity observed at 24 h of treatment could be attributed to aggregation of the PS, resulting in a decrease in fluorescence. Previous reports have documented that both monomer and aggregated forms of PSs can be present in cells, where the aggregated state exhibits diminished fluorescence and photosensitizing efficiency [48,49]. The PS moiety used in CTSB -PPP is reported to exhibit significant dark toxicity on normal Vero cells at 10 μM concentration, where the viability was reduced to 43% [19]. Therefore, we assessed whether the coupling of Pha to a polymeric chain could influence dark toxicity. Our study corroborates the Vero cell assay, as CTSB-PPP showed cytotoxicity at 12 µM. Nevertheless, CTSB-PPP was safe at 6 µM concentration (Pha equivalent).

Ex vivo toxicity provided further confirmation of the reliable safety of CTSB-PPP. Vascular function assays with isolated tissue baths evaluated the potential deleterious effect on aortic rings isolated from wild type mice. The primary advantage of this technique is that the tissue is living and functions as a whole tissue, with a physiological outcome (contraction or relaxation) that is relevant to the body [50]. Our results showed that vascular integrity was maintained, and CTSB had no deleterious effect on maximum tissue contractility and endothelium-dependent relaxation. Imaging of protease activity has been effectively utilized for in vivo diagnostic imaging of tumors, arthritis [51], cancer [52], and vulnerable atherosclerotic plaques [53]. Pheophorbide-a, in CTSB-PPP, is highly fluorescent in the near-infrared (NIR) spectrum. NIR shows lower photodamage effects, greater tissue penetration depth, and a higher signal-to-noise ratio than UV or visible light. Thus, the localized accumulation of CTSB-PPP in tissues with high protease activity, along with the fluorescence of pheophorbide-a, can be used in cancer bioimaging techniques. Altogether, our results suggest that CTSB-PPP may be a good candidate as a diagnostic tool due for the distinct expression profiling of CTSB in disease pathology.

PDT efficacy using CTSB-PPP was demonstrated in vitro. Significant reduction in cell viability upon photo-illumination showed an adequate photodynamic response. The active pheophorbide-a exerts cytotoxicity by several mechanisms, mainly based on apoptosis and the autophagy pathway [18]. Pha-mediated PDT is known to upregulate the expression of BCL-2 (B-cell lymphoma-2), BAX (Bcl-2-associated X protein), Caspase-3, and PARP (poly adenosine diphosphate-ribose polymerase) [54]. In addition, it is also reported that pheophorbide-a based PDT reduces the phosphorylation of mTOR, which is involved in cancer cell proliferation [55]. These reports suggest that CTSB-PPP-based PDT may promote the signaling involved in cell apoptosis and inhibit the activation of mTOR to suppress cell proliferation.

Upon illumination, PS molecules generate hydroxyl radical, superoxide anion, hydrogen peroxide, and singlet oxygen. Singlet oxygen can further react with nearby molecules to induce the formation of other ROS species that can be detected via chemical sensors, such as DHE [39,56]. When ROS concentration surpasses a certain level, it leads to apoptosis via the loss of mitochondrial membrane potential, lipid peroxidation, or the protein denaturation of cellular membranes and organelles [12,13]. Our results demonstrate that CTSB-PPP in the absence of light does not induce ROS production, again confirming the absence of dark toxicity of the PS at the selected dose. The photo-illumination of CTSB-PPP led to a ~4-fold increase in ROS level, attributed to its efficient phototoxic effects.

## 5. Conclusions

CTSB-PPP showed no dark toxicity at low concentrations. This probe may thus be utilized as a potential imaging agent to identify cells or tissues with high cathepsin B protease activity. CTSB-PPP-based PDT results in effective cytotoxicity and therefore, holds promise as a therapeutic agent for achieving the highly selective destruction of cells with high proteolytic activity, for instance, in neoplasms. However, a more detailed in vivo study is required to establish proof of this concept for the CTSB-PPP-based selective targeting of cathepsin B.

## Figures and Tables

**Figure 1 pharmaceuticals-15-00564-f001:**
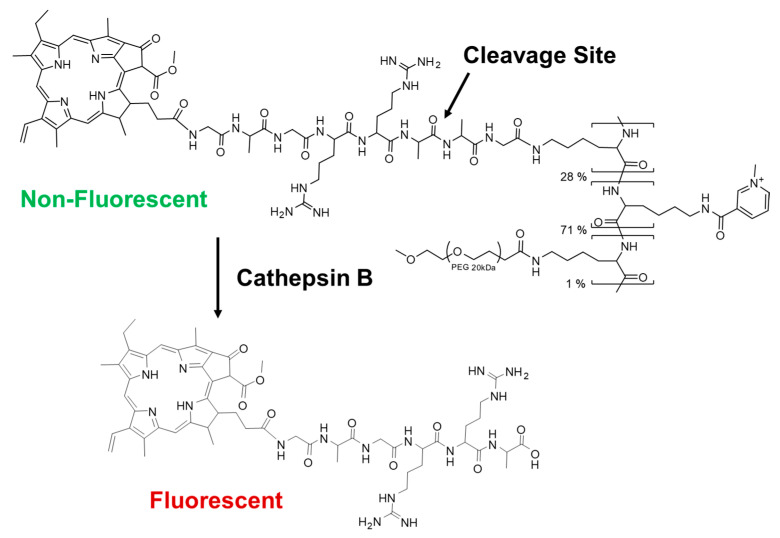
Structural representation of CTSB-PPP and the principle of enzyme mediated activation of CTSB-PPP. The prodrug is synthesized with an average loading of 28% Pha-peptide per polymer chain. Intramolecular interactions occur between closely positioned PSs in the intact prodrug, resulting in reduced fluorescence emission. Cathepsin B-mediated cleavage of the peptide linker leads to the release of PS-peptidyl fragments, which are again fully photoactive.

**Figure 2 pharmaceuticals-15-00564-f002:**
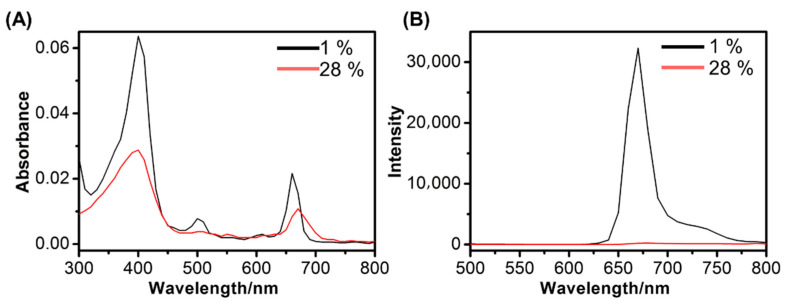
Fluorescence spectra of CTSB-PPP. (**A**) absorption spectra and (**B**) Emission spectra of CTSB-PPP (28% loading of Pha-peptide) and its non-quenched reference (1% loading of Pha-peptide) with equimolar amount of Pha (3.0 µM) in aqueous DPBS buffer. Ex = 400 nm; Em = 670 nm; T = 37 °C.

**Figure 3 pharmaceuticals-15-00564-f003:**
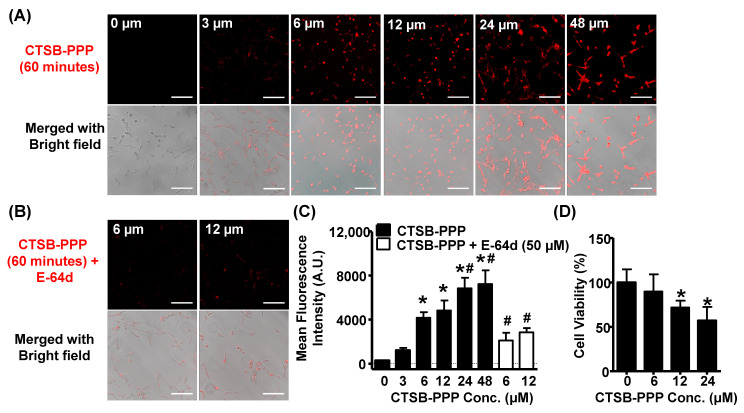
Dose-dependent cellular accumulation of CTSB-PPP. (**A**,**B**) BMCs were treated for 1 h with CTSB-PPP at different concentrations, in the presence or absence of E-64d, and intracellular accumulation was monitored by confocal microscopy. The upper panel illustrates characteristic fluorescence of cleaved PS consisting of Pha as fluorophore in red, and the lower panel is merged with corresponding bright field images, illustrating localization. Scale bar: 100 µm. (**C**) Bar diagram representing quantification of fluorescence intensity per cell in each group (*n* = 3), and (**D**) BMCs were treated with varying concentration of CTSB-PPP for 1 h. Cell viability was detected by MTT assay after 24 h of treatment (*n* = 3). * *p* < 0.05 vs. vehicle treated i.e., 0 µM CTSB-PPP treated cells; # *p* < 0.05 vs. 6 or 12 µM CTSB-PPP treated cells. Statistical analysis—one-way ANOVA, followed by Bonferroni’s post hoc test.

**Figure 4 pharmaceuticals-15-00564-f004:**
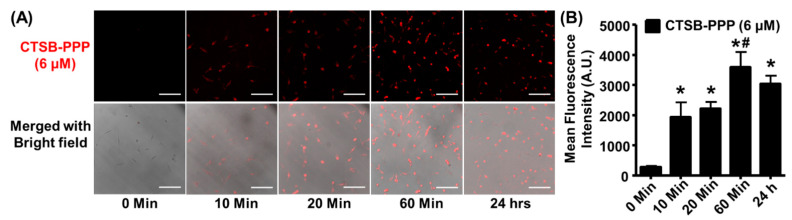
Time-dependent activation and accumulation of CTSB-PPP. (**A**) Confocal fluorescence micrographs merged with corresponding bright field images of BMCs treated with 6 µM CTSB-PPP for 10 min, 20 min, 60 min, or 24 h. Scale bar: 100 µm. (**B**) Bar graph representing the fluorescence intensity per cell in each group (*n* = 3). * *p* < 0.05 vs. 10 min, # *p* < 0.05 vs. 20 min. Statistical analysis—one-way ANOVA, followed by Bonferroni’s post hoc test.

**Figure 5 pharmaceuticals-15-00564-f005:**
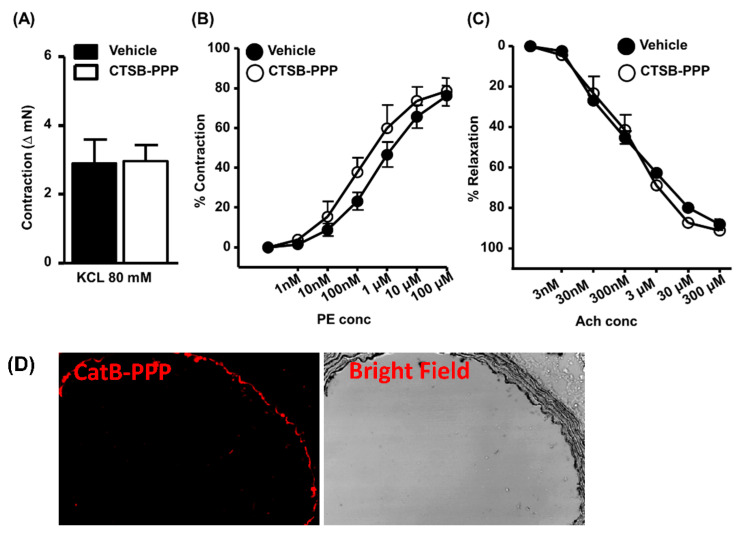
CTSB-PPP itself does not affect vascular function. Aortic segments from C57BL/6J mice were mounted in Myograph. Vascular function was assessed before or after CTSB-PPP (6 µM for 1 h) treatment (**A**) Bar diagram representing KCl induced contraction, (**B**) Dose-dependent phenylephrine (1 Nm–100 μM) induced contractions, Scale bar: 100 µm, and (**C**) Acetylcholine (3 nM–300 µM) mediated relaxations (*n* = 4). Statistical analysis—unpaired two-tailed Student’s *t*-test. (**D**) CTSB-PPP (6 µM for 1 h) was perfused ex vivo through the thoracic aorta isolated from wild type C57Bl/6J mice. Representative images showing characteristic fluorescence of cleaved PS consisting of Pha as fluorophore in red, along with corresponding bright field images of aortic sections.

**Figure 6 pharmaceuticals-15-00564-f006:**
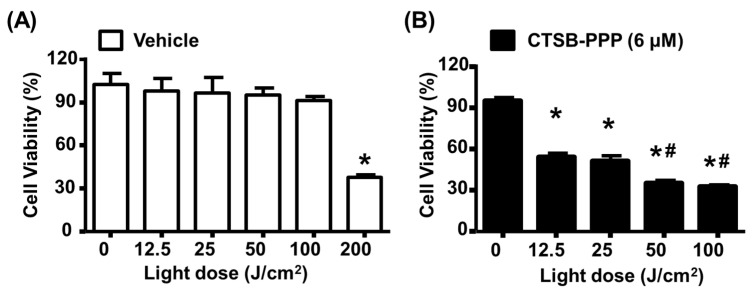
PDT-induced photo toxicity. BMCs were treated, with or without CTSB-PPP (6 µM), for 1 h. PDT was applied immediately after treatment. Cell viability was determined after 24 h by MTT assay. Bar diagram representing (**A**) effect of different dose of laser irradiation on the viability of BMCs and (**B**) phototoxicity of CTSB-PPP illuminated with red light at fluence of 12.5 J/cm^2^ (*n* = 3). * *p* < 0.05 vs. vehicle treated non-illuminated control cells. # *p* < 0.05 vs. 12.5 or 25 J/cm^2^ treated cells. Statistical analysis—one-way ANOVA, followed by Bonferroni’s post hoc test.

**Figure 7 pharmaceuticals-15-00564-f007:**
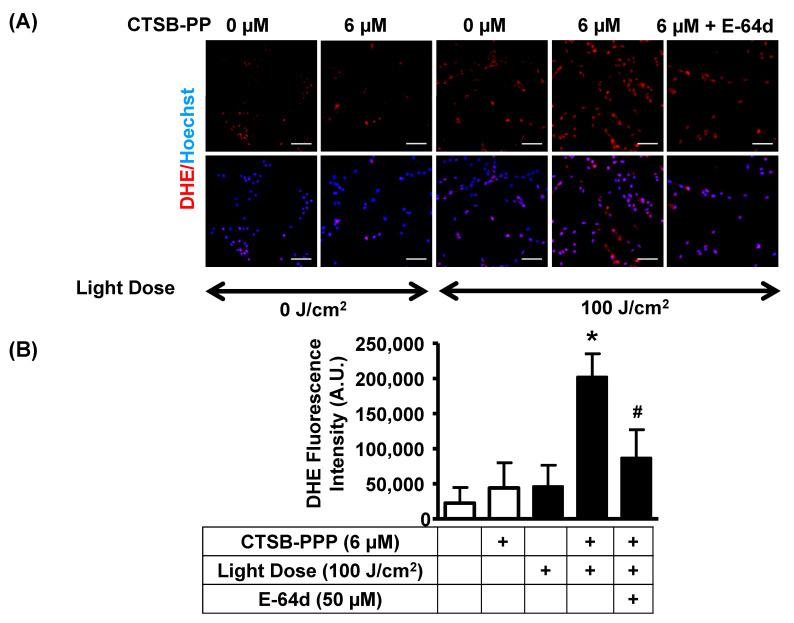
ROS production by CTSB-PPP-mediated PDT. ROS levels were determined immediately after PDT by DHE staining (**A**) Upper-top panel show a representative fluorescent photomicrograph of DHE staining, indicating ROS signals as red fluorescence, and upper-bottom panel represents the merged image with Hoechst in blue. Scale bar: 100 µm. (**B**) Bar diagram representing DHE fluorescence intensity/µm^2^. * *p* < 0.05 vs. vehicle-treated, un-illuminated control. # *p* < 0.05 vs. CTSB-PPP-treated photo-illuminated cells. Statistical analysis—one-way ANOVA, followed by Bonferroni’s post hoc test.

## Data Availability

All data is available in this manuscript and its accompanying supplementary information.

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
