# Peer review of "Cathepsin B-Cleavable Polymeric Photosensitizer Prodrug for Selective Photodynamic Therapy: In Vitro Studies"

_pharmaceuticals, 2022, doi:10.3390/ph15050564_

Round 1

Reviewer 1 Report

In this work, the authors reported a Cathepsin-B cleavable polymeric photosensitizer prodrug (CTSB-PPP) for selective photodynamic therapy against CTSB overexpressed tumor cells. Some specific should be addressed to confirm the work.

  1. In Figure 2, the UV-vis absorption and fluorescence emission of CTSB-PPP was evaluated with different grafting degree of PPa. However, the the fluorescence emission of CTSB-PPP was suggested to be evaluated to confirm the CTSB responsibility.
  2. A negative control of CTSB expressed cell line should be used to evaluate the fluorescence recovery and cytotoxicity to confirm the intracellular activation and selectivity of CTSB-PPP.
  3. The ROS production ability of CTSB-PPP should be evaluated in cuvette.
  4. The characterizations of CTSB-PPP should be provided.

Reviewer 2 Report

Comments: The work is an interesting manuscript,  However there are few conceptual issues in this article, Would you please clarify following arguments?

  • The cytotoxicity of CTSB-PPP was investigated by using isolated bone marrow cells because of their expression of Cathepsin B. However, other normal cells and cancer cells are strongly recommended to investigate drug efficacy and cytotoxicity.
  • MTT assay revealed that there was growth reduction  for normal  BMCs at 24 μM by 57 %.  However, authors did not discuss how much is it safe to use it later in  human cancer  therapy? Or how to protect normal cells from its cytotoxicity.
  • Authors did not provide clear experiment that Cathepsin B is responsible directly for cleaving the linker of CTSB-PPP. In vitro experiment  (dialysis technique) should to be used for detecting the activation of Cathepsin B incubated with CTSB-PPP.
  • Authors did not provide clear explanation for why CTSB-PPP can be used as pro drug? Which is signalling pathway can be blocked? Why is it toxic inside cytoplasm ? how its fluorescence intensity can be useful in cancer bio image techniques ? and what its behaviour after  using  in nano formulation ?
  • Three characteristic peaks of CTSB-PPP were exhibited at Fig 2 around 400 nm, 500 nm and 650 nm. However, one peak was emitted at around 657 nm and its intensity is very low. Authors did not explain this matter according to physical properties of CTSB-PPP.  Additionally, the  fluorescence intensity was measured at 500 nm to 800 nm. authors have to measure this intensity from 300nm to 800 nm. another peak could be appeared  at around 490 nm  and it will emit FITC color.  Authors should to clarify this condition.
  • Authors did not explain why CTSB-PPP was accumulated inside cytoplasm through 1 h ? which factors could facilitate its adsorption?

Reviewer 3 Report

Proteases, including intracellular proteases, play an important role in many stages of tumor growth. High levels of cathepsin B, a lysosomal cysteine ​​cathepsin, are known to be found in many human cancers, levels that often cause cathepsin B to be secreted and associated with the tumor cell membrane. In experimental models, such as transgenic mouse pancreatic and mammary carcinoma models, cathepsin B has been shown to play a fundamental role in tumor cell initiation, growth/proliferation, angiogenesis, invasion, and metastasis. In transgenic models, the absence of cathepsin B has been associated with increased apoptosis, but cathepsin B has also been shown to promote apoptosis. Cathepsin B is part of a proteolytic pathway identified in human glioma xenograft models; targeting cathepsin B alone in these tumors is less effective than targeting cathepsin B in combination with other proteases or protease receptors. In connection with the foregoing, studies on the study of Cathepsin B derivatives as polymeric photosensitizers for selective photodynamic therapy are of significant interest and can be published in Pharmaceuticals.
In the meantime, I have a few comments and suggestions for the discretion of the authors:
- In the water part of the article, it is necessary to discuss in more detail the known drugs for photodynamic therapy, their pros and cons. In addition, it is not entirely clear what is the reference substance. And how do the data in the proposed article compare with similar drugs known in the literature and used today in medical practice?
- Is it possible to study the induction of apoptosis and the effect on the cell cycle for the test compound using flow cytometry?
- ex-vivo, in vivo – should be in Italiс in the text.

Round 2

Reviewer 2 Report

Authors revised  manuscript point by point  according to comment of reviewer. Manuscript is more acceptable NOW.